

# Isolation and sequence-based characterization of a koala symbiont: *Lonepinella koalarum*

Katherine E. Dahlhausen[1], Guillaume Jospin[1], David A. Coil[1], Jonathan A. Eisen[1,2,3] and Laetitia G.E. Wilkins[1]

[1] Genome and Biomedical Sciences Facility, University of California, Davis, Davis, CA, USA
[2] Department of Evolution and Ecology, University of California, Davis, Davis, CA, USA
[3] Department of Medical Microbiology and Immunology, University of California, Davis, Davis, CA, USA

Corresponding author
Laetitia G.E. Wilkins,
megaptera.helvetiae@gmail.com

## ABSTRACT

Koalas (*Phascolarctos cinereus*) are highly specialized herbivorous marsupials that feed almost exclusively on *Eucalyptus* leaves, which are known to contain varying concentrations of many different toxic chemical compounds. The literature suggests that *Lonepinella koalarum*, a bacterium in the Pasteurellaceae family, can break down some of these toxic chemical compounds. Furthermore, in a previous study, we identified *L. koalarum* as the most predictive taxon of koala survival during antibiotic treatment. Therefore, we believe that this bacterium may be important for koala health. Here, we isolated a strain of *L. koalarum* from a healthy koala female and sequenced its genome using a combination of short-read and long-read sequencing. We placed the genome assembly into a phylogenetic tree based on 120 genome markers using the Genome Taxonomy Database (GTDB), which currently does not include any *L. koalarum* assemblies. Our genome assembly fell in the middle of a group of *Haemophilus*, *Pasteurella* and *Basfia* species. According to average nucleotide identity and a 16S rRNA gene tree, the closest relative of our isolate is *L. koalarum* strain Y17189. Then, we annotated the gene sequences and compared them to 55 closely related, publicly available genomes. Several genes that are known to be involved in carbohydrate metabolism could exclusively be found in *L. koalarum* relative to the other taxa in the pangenome, including glycoside hydrolase families GH2, GH31, GH32, GH43 and GH77. Among the predicted genes of *L. koalarum* were 79 candidates putatively involved in the degradation of plant secondary metabolites. Additionally, several genes coding for amino acid variants were found that had been shown to confer antibiotic resistance in other bacterial species against pulvomycin, beta-lactam antibiotics and the antibiotic efflux pump KpnH.
In summary, this genetic characterization allows us to build hypotheses to explore the potentially beneficial role that *L. koalarum* might play in the koala intestinal microbiome. Characterizing and understanding beneficial symbionts at the whole genome level is important for the development of anti- and probiotic treatments for koalas, a highly threatened species due to habitat loss, wildfires, and high prevalence of *Chlamydia* infections.

# INTRODUCTION

Koalas (*Phascolarctos cinereus*) are arboreal marsupials that are highly specialized herbivores in that they feed almost exclusively on the foliage of select *Eucalyptus* species (*Moore & Foley, 2005*; *Callaghan et al., 2011*). All *Eucalyptus* species contain chemical defenses against herbivory that include tannins, B-ring flavanones, phenolic compounds, terpenes, formylated phloroglucinols, cyanogenic glucosides and other plant secondary metabolites (*Lawler, Foley & Eschler, 2000*; *Gleadow et al., 2008*; *Brice et al., 2019*; *Liu et al., 2019*). Plant chemical defenses can deter herbivores by affecting taste and/or digestibility of ingested material, with varying levels of toxic effects (*Brice et al., 2019*). These defenses and anti-nutrient compounds, which will be referred to generally as plant chemical defenses (for "PCDs") hereafter, are very common, and there is well-established evidence that herbivores can overcome these defenses in their diet at least in part via PCD-degradation by their intestinal microbial communities (*Freeland & Janzen, 1974*; *Waterman et al., 1980*; *Hammer & Bowers, 2015*; *Kohl & Denise Dearing, 2016*). However, it is unknown to what extent *Eucalyptus* PCDs are degraded by the intestinal microbial communities of koalas.

Research has highlighted several ways in which koalas are able to manage such a toxic diet independent of the functions of their intestinal microbial communities. For example, some studies suggest that koalas can minimize PCDs intake through tree- and even leaf selection (*Lawler, Foley & Eschler, 2000*; *Marsh et al., 2003*; *Moore & Foley, 2005*; *Liu et al., 2019*). Another study identified several genes in the koala genome that are associated with metabolism and detoxification of many types of xenobiotics (*Johnson et al., 2018*). The findings from a study on microsomal samples from koala liver also suggest that koalas are able to metabolize some xenobiotics in their livers (*Ngo et al., 2000*). Furthermore, other research suggests that toxic compounds found in *Eucalyptus* may be absorbed in the upper digestive system before they even reach the intestinal microbial community of herbivores (*Foley, Lassak & Brophy, 1987*).

The intestinal microbial communities of koalas are also thought to contribute to the management and degradation of PCDs found in *Eucalyptus* leaves. To date, the most compelling evidence for this is included in a recent study on the koala fecal microbiome, which identified several metabolic pathways and relevant bacterial species proposed to be important in detoxification processes (*Shiffman et al., 2017*). The koala microbiome as a whole has shown to play an important role in macro nutrient digestion and fiber degradation (*Blyton et al., 2019*; *Brice et al., 2019*). Moreover, there is evidence that the koala gastrointestinal microbiome can influence diet selection of individual hosts (*Blyton et al., 2019*). At the individual level, several bacterial isolates associated with the intestinal microbial communities of koalas have been characterized in the context of degradation of PCDs found in *Eucalyptus* leaves (*Osawa, 1990*, *1992*; *Osawa et al., 1993*, *1995*;

*Looft, Levine & Stanton, 2013*). One of these cultured isolates is a bacterium known as *Lonepinella koalarum*. It had been first isolated from the mucus around the caecum in koalas and was shown to degrade tannin-protein complexes that can be found in *Eucalyptus* leaves (*Osawa et al., 1995*; *Goel et al., 2005*). Briefly, tannin-protein complexes are extremely diverse and result from the reaction between plant defense secondary metabolites; i.e., tannins, and proteins. Tannins bind proteins followed by the formation of a precipitate, which cannot be digested by koalas or utilized by microbes (*Adamczyk et al., 2017*). In our previous work, *L. koalarum* was identified as the most predictive taxon of koala survival during antibiotic treatment (*Dahlhausen et al., 2018*). Briefly, a co-occurrence network analysis identified four bacterial taxa, including one of the genus *Lonepinella*, that could be found in feces of koalas that survived their antibiotic treatment after *Chlamydia* infection. However, these four taxa were absent from feces of koalas that died. Furthermore, in the same study a random forest analysis revealed that the most predictive taxon of whether a koala would live or die during their antibiotic treatment was identified as *L. koalarum*. This finding suggests that *L. koalarum* could be important for koala health, but the study did not present any evidence relating to PCD degradation in the highly specialized diet of koalas.

It is well understood that animals with highly specialized diets also are likely to have highly specialized intestinal microbial communities (*Higgins et al., 2011*; *Kohl et al., 2014*; *Alfano et al., 2015*; *Kohl, Stengel & Denise Dearing, 2016*). Disturbances of a specialized microbial community, such as the introduction of antibiotics, can have profound effects on the host's health (*Kohl & Denise Dearing, 2016*; *Brice et al., 2019*). Yet, koalas are regularly treated with antibiotics due to the high prevalence of *Chlamydia* infections in many populations (*Polkinghorne, Hanger & Timms, 2013*). While recent advances in *Chlamydia pecorum* vaccines for koalas are a promising alternative for managing koala populations, antibiotics are still the current treatment method for bacterial infections in koalas (*Waugh et al., 2016*; *Desclozeaux et al., 2017*; *Nyari et al., 2018*). The antibiotics used in practice might not only target *Chlamydia pecorum* but also beneficial koala gut symbionts as a side effect. Therefore, it is important to learn about bacteria associated with koala health, such as *L. koalarum*, in order to further the development of alternative treatments for bacterial infections in koalas and to recommend antibiotic compounds that are potentially less disruptive to members of the koala gut microbiome.

Here we isolated a strain of *L. koalarum* (hereafter called strain UCD-LQP1) from the feces of a healthy koala (*P. cinereus*) female at the San Francisco Zoo. We sequenced the genome of *L. koalarum* UCD-LQP1 using a combination of long- and short-read sequencing, and then assembled and annotated the genome. We compared the genome assembly to the most closely related genomes that are currently publicly available. The genome assembly of *L. koalarum* UCD-LQP1 was placed in a phylogenetic tree and screened for genes putatively involved in the degradation of plant secondary metabolites, carbohydrate metabolism, and antibiotic resistance. Additionally, we identified and characterized putative genes that were unique to this strain and two recently sequenced genomes of *L. koalarum* from Australia.

## MATERIALS AND METHODS

### Sampling of koala feces and preparation of culturing media

A koala fecal pellet was collected, with permission from the San Francisco Zoo, from a healthy, adult, captive, female koala (*Phascolarctos cinereus*). We do not have any information on the geographical origin of this koala. Koalas at the SF Zoo are fed blue gum leaves (*Eucalyptus globulus*), which grow quite abundantly in California. Jim Nappi and Graham Crawford of the San Francisco Zoo organized and permitted koala fecal sample collection. The fresh fecal pellet was collected from the floor with sterilized tweezers and stored in a sterile 15 ml Falcon tube (Thermo Fisher Scientific, Waltham, MA, USA). The tube was immediately placed on ice after collection and subsequently stored at 4 °C overnight.

The preparation of the *Lonepinella koalarum* culturing media was modified from methods developed by Osawa et al. (1995). A 2% agarose (Fisher BioReagents, Waltham, MA, USA) solution of Bacto™ Brain Heart Infusion (BHI; BD Biosciences, Franklin Lakes, NJ, USA) was prepared following manufacturer protocols. After the media had solidified in petri dishes, a 2% tannic acid solution was prepared by combining 1 g of tannic acid powder per 50 ml of sterile Nanopure™ water (Spectrum Chemical MFG CORP, New Brunswick, NJ, USA). The solution was vortexed for 1 min until homogenized, resulting in a brown, transparent liquid. Using a sterile serological pipette, 5 ml of the 2% tannic acid solution was gently added to each BHI media plate and left for 20 min. After incubation, the remaining liquid on the plate was decanted. No antibiotic compounds were added to the medium.

### Culturing of isolates and DNA extraction

The koala fecal pellet was cut in half with sterile tweezers. Tweezers were re-sterilized and used to move approximately 300 mg of material from the center of the pellet to a sterile 2 ml Eppendorf tube containing 1 ml of sterile, Nanopure™ water. The tube was vortexed for 3 min, intermittently checking until the solution was homogenized into a slurry. One hundred µl of the homogenized fecal slurry was micro-pipetted onto to a BHI +tannin plate and stored in an anaerobic chamber (BD GasPak™ EZ anaerobe chamber system; BD Biosciences, Franklin Lakes, NJ, USA) at 37 °C for 3 days. Each individual colony that grew was plated onto a freshly made BHI+tannin plate using standard dilution streaking techniques. The new plates were stored in an anaerobic chamber at 37 °C for another 3 days. This step was repeated two more times to decrease the probability of contamination or co-culture.

An individual colony from each of the plates from the third round of dilution streaking was moved to a sterile 30 ml glass culture tube containing 5 ml of sterile Bacto™ BHI liquid media (prepared following manufacturer protocol; BD Biosciences, Franklin Lakes, NJ, USA). Each tube was then capped with a sterile rubber stopper and purged with nitrogen gas in order to create an anaerobic environment. The tubes were placed in an incubated orbital shaker (MaxQ™ 4450; ThermoFisher Scientific, Waltham, MA, USA) for 3 days at 37 °C at 250 rpm.

Using a sterile serological pipette, 1.8 ml of each liquid culture was transferred to a sterile 2 ml Eppendorf tube. The tubes were spun at 13,000 g for 2 min and the supernatant was carefully decanted. The DNA was extracted from the pellet in each sample with the Promega Wizard Genomic DNA Purification Kit (Promega, Madison, WI, USA) according to the manufacturer's protocol. DNA was eluted in a final volume of 100 µl and stored at 4 °C.

## PCR and sanger sequencing

PCR amplification of the 16S rRNA gene was performed on each of the eluted DNA samples. PCR reactions were prepared using the bacteria-specific "universal" primer pair 27F (5′-AGAGTTTGATCMTGGCTCAG-3′; *Stackebrandt & Goodfellow, 1991*) and 1391R (5′-GACGGGCGGTGTGTRCA-3′; *Turner et al., 1999*). PCR amplifications were performed in a BioRad T100™ Thermal Cycler in 50 µl reactions. Each reaction contained 2 µl of the eluted DNA from the aforementioned extraction, 5 µl of 10x Taq buffer (Qiagen, Valencia, CA, USA), 10 µl of Q buffer (Qiagen, Valencia, CA, USA), 1.25 µl of 10mM dNTPs (Qiagen, Valencia, CA, USA), 2.5 µl of 10mM 27F primer, 2.5 µl of 10mM 1391R primer, 0.3 µl of Taq polymerase (Qiagen, Valencia, CA, USA), and 26.45 µl of sterile water. The cycling conditions were: (1) 95 °C for 3 min, (2) 40 cycles of 15 s at 95 °C, 30 s at 54 °C, and 1 min at 72 °C, (3) a final incubation at 72 °C for 5 min and (4) holding at 12 °C upon completion.

The PCR product for each sample was purified and concentrated by following the manufacturer's protocol for the NucleoSpin Gel and PCR Clean-up kit (Macherey-Nagel, USA). The purified PCR product for each sample was quantified by following the manufacturer's protocol for the Qubit dsDNA HS Assay Kit (Thermo Fisher Scientific, Waltham, MA, USA). The PCR product for each sample was then diluted to 26 ng/µl and submitted for forward and reverse Sanger sequencing at the University of California Davis DNA Sequencing Facility. The program SeqTrace version 0.9.0 (*Stucky, 2012*) was used to edit and create consensus sequences of the reads received from the sequencing facility, following the protocol detailed in *Dunitz et al. (2015)*. The consensus sequence for each sample was uploaded to the NCBI blast website for organism identification (*Madden, 2003*). The DNA of one of the isolates that had been identified as *L. koalarum* was used for whole-genome sequencing, as described below. We refer to this isolate as *L. koalarum* strain UCD-LQP1.

## Whole genome sequencing and assembly

DNA from one sample identified as *L. koalarum* strain UCD-LQP1 was submitted for whole genome PacBio sequencing at SNPsaurus | GENOMES to GENOTYPES (https://www.snpsaurus.com). After sequencing, the demultiplexed bam file was tested for reads that contained palindromic sequences since a preliminary assembly with Canu version 1.8 (*Koren et al., 2017*) indicated the presence of adapter sequences. Palindromic reads were split in half, aligned with minimap2 (an executable in Canu), and those palindromic reads that aligned over at least two-thirds of the split read were reduced to the

first part of the palindrome (*Koren et al., 2017*). This procedure efficiently removed adapter sequences. These adapter-free reads were used in the hybrid assembly described below.

The same DNA that had been used for PacBio sequencing was also submitted for Illumina sequencing. Ten ng of genomic DNA were used in a 1:10 reaction of the Nextera DNA Flex Library preparation protocol (Illumina, San Diego, CA, USA). Fragmented DNA was amplified with Phusion DNA polymerase (New England Biolabs, Ipswich, MA, USA) in 12 PCR cycles with 1 min extension time. Samples were sequenced on a HiSeq4000 instrument (University of Oregon GC3F) with paired-end 150 bp reads. The 10,309,488 raw reads were quality controlled and filtered for adaptors and PhiX using the BBDuk tool version 37.68 (*Bushnell, 2014*), resulting in 10,302,312 reads. The 308 cleaned PacBio reads and 10,302,312 filtered Illumina reads were combined with all default parameters of Unicycler version 0.4.5, a tool used to assemble bacterial genomes from both long and short reads (*Wick et al., 2017*).

## Genome annotation

Completeness and contamination of the *L. koalarum* strain UCD-LQP1 assembly were determined with CheckM version 1.0.8 (*Parks et al., 2015*), number of contigs, total length, GC%, N50, N75, L50, and L75 were determined with QUAST (Quality Assessment Tool for Genome Assemblies; *Gurevich et al., 2013*), and the assembly was annotated with PROKKA version 1.12 (*Seemann, 2014*). The *L. koalarum* strain UCD-LQP1 genome assembly was uploaded to the Rapid Annotation using Subsystem Technology online tool (RAST), a genome annotation program for bacterial and archaeal genomes (*Aziz et al., 2008*). The SEED viewer in RAST was used to browse features of the genome (*Overbeek et al., 2014*). To screen the *L. koalarum* strain UCD-LQP1 assembly for genes putatively involved in tannin degradation and xenobiotic metabolisms; *i.e.*, the degradation of plant secondary metabolites, coding regions in the assembly were identified using Prodigal version 2.6.3 (*Hyatt et al., 2010*). Each identified coding region was annotated using eggNOG (a database of orthologous groups and functional annotation that is updated more regularly than PROKKA) mapper version 4.5.1 (*Jensen et al., 2008*). Then, KEGG (Kyoto Encyclopedia of Genes and Genomes) pathways putatively involved in xenobiotics biodegradation and metabolism, were extracted from the eggNOG annotations (Class 1.11 Xenobiotics biodegradation and metabolism includes the following KEGG pathways: ko00362, ko00627, ko00364, ko00625, ko00361, ko00623, ko00622, ko00633, ko00642, ko00643, ko00791, ko00930, ko00363, ko00621, ko00626, ko00624, ko00365, ko00984, ko00980, ko00982 and ko00983 (*Kanehisa & Goto, 2000*)), and the corresponding nucleotide sequences from the *L. koalarum* genome assemblies were saved. Individual genes with hits in KEGG pathways were manually mapped onto KEGG reference maps using the KEGG webtool (*Kanehisa & Goto, 2000*).

## 16S rRNA gene based phylogenetic placement of genome

The 16S rRNA gene sequence within the genome assembly was extracted from RAST by searching for "ssu rRNA" in the function search of the SEED genome browser (*Aziz et al., 2008*; *Overbeek et al., 2014*). Following the protocol outlined in *Dunitz et al. (2015)*,

the 16S rRNA gene sequence was uploaded to the Ribosomal Database Project (RDP; *Cole et al., 2014*) and grouped with all 16S rRNA gene sequences in the Pasteurellaceae family and one chosen outgroup, *Agarivoran* spp., to root the tree. The taxon names from the RDP output file were manually cleaned up and their 16S rRNA gene sequences were used to build a phylogenetic tree with the program FastTree (*Price, Dehal & Arkin, 2009*). Nodes and tip labels were manually edited for Fig. 1 in iTOL (interactive tree of life; web tool; *Letunic & Bork, 2019*). The 16S rRNA gene sequence alignment (*Wilkins & Coil, 2020a*) and its resulting phylogenetic tree are available on Figshare (*Wilkins & Coil, 2020b*). During the preparation of this manuscript, two more *L. koalarum* type strains had their genomes sequenced: one by the DOE Joint Genome Institute, USA (GenBank accession number GCF_004339625.1; 2,486,773 bp long) and one by the Maclean Lab in Australia (GenBank accession number GCF_004565475.1; 2,509,358 bp). Both assemblies were based on type strains originating from the same isolation of *L. koalarum* in 1995 (*Osawa et al., 1995*), DSM 10053 and ATCC 700131, respectively. These two *L. koalarum* genome assemblies were henceforth included in our analysis. When we refer to all three *L. koalarum* genome assemblies, we simply say 'in *L. koalarum*' and when we refer to the strain sequenced in this study, we use 'the assembly of *L. koalarum* strain UCD-LQP1'.

## Comparative genomics

The GTDB-Tk software toolkit version 0.3.0 (*Chaumeil, Hugenholtz & Parks, 2018*) of the Genome Taxonomy Database (GTDB) project was chosen to place *L. koalarum* into a pre-generated conserved marker gene tree using 120 marker genes (*Parks et al., 2018*). After placing the assembly into the GTDB tree, a clade in the tree was extracted that contained *L. koalarum* strain UCD-LQP1 and 55 other taxa, of which all members belonged to the order Pasteurellales. This clade contained all sequenced genomes of the closest neighboring taxa ($n = 55$) to *L. koalarum* in the GTDB tree at the time of this analysis (3 August 2019). All of these 55 genomes were downloaded from GenBank (using the accession numbers in the GTDB) to perform a comparative genomic analysis in Anvi'o version 5.5 (*Eren et al., 2015*). The two other *L. koalarum* genomes from GenBank were included in the following analysis as well. Accession numbers of all genome assemblies included can be found in Table S1 ($n = 58$). The Anvi'o workflow for microbial pangenomics was followed (*Delmont & Eren, 2018*). The blastp program from NCBI was used for a gene search (*Altschul et al., 1990*), the Markov Cluster algorithm (MCL) version 14.137 (*Van Dongen & Abreu-Goodger, 2012*) was used for clustering, and the program MUSCLE was used for alignment (*Edgar, 2004*). An inflation parameter of 6 was chosen to identify clusters in amino acid sequences. Genomes in the pangenome of Anvi'o were ordered based on a genomic marker gene tree. This tree was built in PhyloSift version 1.0.1 (*Darling et al., 2014*) with its updated markers database (version 4, posted on 12th of February 2018; *Jospin, 2018*) for the alignment. We used RAxML version 8.2.10 on the CIPRES web server for the tree inference (*Miller, Pfeiffer & Schwartz, 2010*) following the analysis in *Wilkins et al. (2019)*. Gene clusters in Anvi'o were ordered based

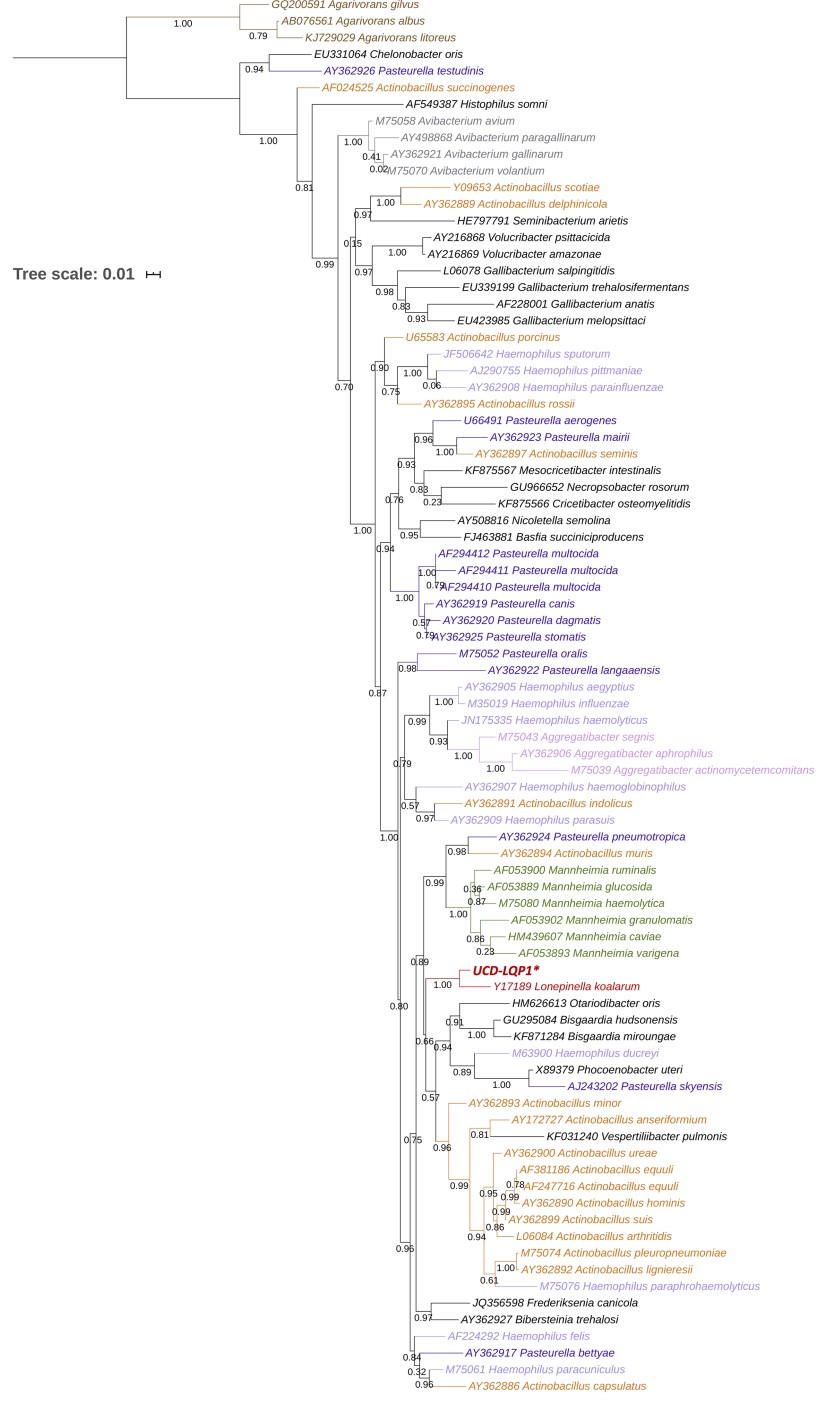

**Figure 1 16S rRNA gene phylogenetic placement of *Lonepinella koalarum* strain UCD-LQP1.** The 16S rRNA gene was extracted from the *L. koalarum* genome assembly by searching for "ssu rRNA" in the RAST function search of the SEED genome browser (*Aziz et al., 2008*; *Overbeek et al., 2014*). Included are all known 16S rRNA sequences in the Pasteurellaceae family and one outgroup, *Agarivoran* spp. Nodes and tip labels are colored corresponding to the Anvi'o profile in Fig. 2; that is, red: *Lonepinella koalarum* (Unicycler: assembly of *L. koalarum* strain UCD-LQP1, in bold and marked with a star), dark purple: *Pasteurella* spp., light purple: *Haemophilus* spp., orange: *Actinobacillus* spp., pink: *Aggregatibacter* spp. and green: *Mannheimia* spp. Black are genera that were not used in Fig. 2, and brown depicts the outgroup *Agarivoran* spp.

on presence/absence. We also used Anvi'o to compute average nucleotide identities across the genomes with PyANI (*Pritchard et al., 2016*). In the heatmap, ANI values > 95% (and >70% for a separate figure, respectively) were colored in red.

Gene clusters from the Anvi'o microbial pangenomics analysis that could only be found in the three *L. koalarum* genome assemblies were extracted. Then, we also extracted all gene clusters that could only be found in the assembly of *L. koalarum* strain UCD-LQP1. Partial sequences were removed. A literature search of the remaining genes was conducted to identify possible roles *L. koalarum* might play in the gut microbiome of koalas. Tables were summarized in R version 3.4.0 (*R Development Core Team, 2013*).

## Carbohydrate metabolism

Since the majority of gene clusters unique to *L. koalarum* genome assemblies fell into the COG (Clusters of Orthologous Groups) category 'Carbohydrate metabolism', we decided to screen all three assemblies against the Carbohydrate-Active Enzymes Database (CAZy), an expert resource for glycogenomics (*Cantarel et al., 2009*; *Lombard et al., 2014*). In brief, CAZy domains were identified based on CAZy family HMMs (Hidden Markov Models) with a coverage of >95% and an *e*-value < 1E−15. Searches were done through dbCAN, a web resource for automated carbohydrate-active enzyme annotation (*Yin et al., 2012*) and CAZy hits were only retained if they had been found with all three search tools. The three search tools included (i) HMMER version 3.3 (*Eddy, 1998*), (ii) DIAMOND version 0.9.29 for fast blast hits in the CAZy database (*Buchfink, Xie & Huson, 2015*; default parameters; that is, *e*-value < 1E−102, hits per query (−*k*) = 1), and (iii) Hotpep version 1 for short, conserved motifs in the PPR (Peptide Pattern Recognition) library (*Busk et al., 2017*; default parameters; that is, frequency > 2.6, hits > 6). For a detailed walk-through of the assembly, annotation, search for KEGG pathways, and comparative genomics analyses, please refer to the associated Jupyter notebook (*Wilkins, 2020a*).

## Identification of antibiotic resistance genes

All three *L. koalarum* genome assemblies were uploaded to the Comprehensive Antibiotic Resistance Database (CARD version 3.0.7; *Jia et al., 2017*) and the ResFinder database version 5.1.0 (*Zankari et al., 2012*) to screen them for putative antibiotic resistance genes and their variants using blastn searches against CARD 2020 reference sequences using default parameters. The Resistance Gene Identifier (RGI) search pipeline was used to detect SNPs (single nucleotide polymorphisms) using the "perfect, strict, complete genes only" criterion on their website. Briefly, antibiotic resistance genes were searched with nucleotide sequences as input. RGI first predicts complete open reading frames (ORFs) using Prodigal version 2.6.3. To find protein homologs in the CARD references, DIAMOND version 0.9.29 is used. The "perfect" algorithm detects perfect matches of individual amino acids to positions in the curated reference sequences that had been previously associated with antibiotic resistance in other bacterial species (*Alcock et al., 2020*).

**Table 1 *Lonepinella koalarum* strain UCD-LQP1 assembly statistics.**

| Statistic | Value |
|---|---|
| Completeness | 99.205 % |
| Contamination | 0.705 % |
| Number of contigs | 29 |
| Total length | 2,608,483 bp |
| GC% | 39.02 |
| N50 | 2,299,135 bp |
| N75 | 2,299,135 bp |
| L50 | 1 |
| L75 | 1 |
| Number of Predicted Genes | 2,551 |
| Number of Protein Coding Genes | 2,479 |

Note:
Completeness and contamination were determined with CheckM version 1.0.8 (*Parks et al., 2015*); number of contigs, total length, GC%, N50, N75, L50, and L75 were determined with QUAST (Quality Assessment Tool for Genome Assemblies) (*Gurevich et al., 2013*); number of predicted genes and number of protein coding genes were determined with PROKKA version 1.12 (*Seemann, 2014*).

## RESULTS AND DISCUSSION

### Identification of isolates

Besides the isolates identified as *L. koalarum*, we had several other colonies growing on the BHI+tannin plates, including isolates with 16S rRNA gene sequences that matched *Bacillus cereus*, *Bacillus nealsonii*, *Bacillus sonorensis* and *Escherichia coli*. *E. coli* was the most common species isolated.

### Assembly taxonomy and gene annotation

The hybrid assembly generated was 2,608,483 bp in length with an N50 of 2,299,135 bp and a coverage of 672. According to the marker gene analysis in CheckM, the assembly was 99.21% complete and less than 1% contaminated with a GC content of 39.02% (see Table 1 for additional details). One contig in the assembly appears to be a 3,899 bp long plasmid. This is indicated by circularity of that contig and positive matches to plasmids in related taxa when uploaded to the NCBI blast website for organism identification (*Madden, 2003*). The two most similar sequences on GenBank were a 71 percent similar sequence of *Pasteurella multocida* strain U-B411 plasmid pCCK411 (accession number FR798946.1) and a 70 percent similar sequence of *Mannheimia haemolytica* strain 48 plasmid pKKM48 (accession number MH316128.1). The putative plasmid sequence was deposited on FigShare (*Wilkins & Jospin, 2020*).

The taxonomy of *L. koalarum* strain UCD-LQP1 was confirmed in three ways. First, a phylogenetic tree was built based on the 16S rRNA gene extracted from the new assembly. This 16S rRNA gene sequence was aligned with other closely related 16S rRNA gene sequences on the RDP website where 16S rRNA gene sequences of type strains are curated and sequences of the closest relatives of a taxon are usually readily available (*Dunitz et al., 2015*). The phylogenetically closest sequence to *L. koalarum* strain UCD-LQP1 in the

16S rRNA gene tree was one from *Lonepinella koalarum* Y17189 (Fig. 1). Second, a whole genome concatenated gene marker tree was inferred using the Genome Taxonomy Database (GTDB), as well as using PhyloSift, in parallel. In the GTDB tree, *L. koalarum* UCD-LQP1 was placed closest to *Actinobacillus succinogenes* (GenBank accession number GCA_000017245.1). Note that as of 3 February 2020, GTDB did not include any of the *L. koalarum* genome assemblies. In the PhyloSift marker gene tree, all three *L. koalarum* assemblies clustered together, and *A. succinogenes* was their phylogenetically closest neighbor (Fig. 2). Third, the average nucleotide identity (ANI) between the genome of the *L. koalarum* type strain (DSM 10053; GenBank accession number GCF_004339625.1) and the assembly of *L. koalarum* UCD-LQP1 was estimated at 98.91 percent (standard deviation 0.17%). The ANI value between *L. koalarum* UCD-LQP1 and GCF_004565475.1 was 98.99 percent (SD 0.15%) and the ANI value between GCF_004339625.1 and GCF_004565475.1 was 99.99 percent (SD 0.08%). Both of these genome assemblies are based on the type strain of *L. koalarum* that originated in 1995 (Osawa et al., 1995). All three approaches confirmed the taxonomy of strain UCD-LQP1 as *Lonepinella koalarum*. Interestingly, *A. succinogenes* (GenBank accession number GCA_000017245.1) belongs now to a different taxonomic group based on GTDB taxonomy, namely *Basfia succinogenes*. *Parks et al. (2018)*, among others (*Hug et al., 2016*; *Castelle & Banfield, 2018*), have suggested relying on whole genome sequencing to reorganize the microbial tree of life, which will result in a majority of changes in classification and naming, and ultimately reflect a more accurate evolutionary relationship among groups.

There were no positive hits for any annotations associated with tannin degradation in the RAST SEED viewer. This negative result is in contrast to the experimentally verified tannin-degrading functions reported for this bacterium (*Osawa et al., 1995*). Moreover, tannic acid powder had been used to prepare the culturing medium and was expected to help select for bacterial tannin degraders. There are several potential explanations for the absence of any positive hits for tannins in the RAST database including (1) the genes responsible for tannin degradation in the assembly of *L. koalarum* UCD-LQP1 are not labeled as such, or (2) *L. koalarum* does not have any tannin-degradation functionality. We thus carried out additional sequence-based analyses searching for possible PCD degrading genes in the new assembly.

According to the annotation with PROKKA, there were 2,551 predicted genes and 2,479 protein coding genes. In comparison, eggNOG predicted 2,370 protein coding genes. Neither annotation included any genes annotated as "tannase". However, among the eggNOG predictions, there were 79 genes putatively involved in Class 1.11 Xenobiotics biodegradation and the degradation of plant secondary metabolites (Table 2). There are 20 KEGG pathways included in this group. We searched for all twenty pathways in the assembly of *L. koalarum* strain UCD-LQP1 and found positive hits in 13 pathways (Table 2). Each hit represents a translated amino acid sequence from the assembly of *L. koalarum* UCD-LQP1 that is encoded by an individual gene in a pathway. The largest proportion of hits (*n* = 15) comprised putative enzymes that are members in this KEGG class, but do not fall into a particular pathway (KEGG pathway ko00983: Drug metabolism—other enzymes). Potential tannin-degrading genes might be found in this

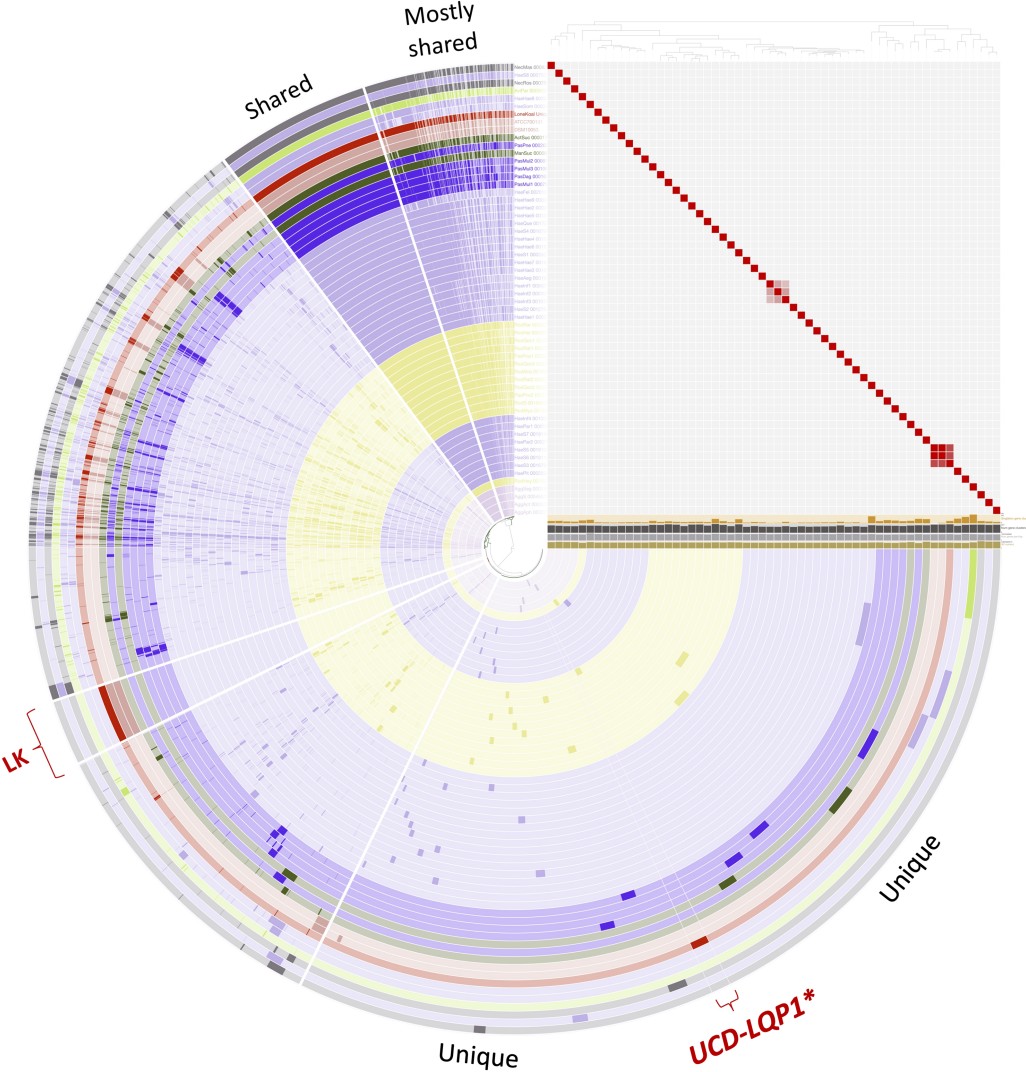

**Figure 2 Pangenome comparison of *L. koalarum* strain UCD-LQP1 and 57 of its most closely related, publicly available genomes.** This figure was generated from the microbial pangenomic analysis in Anvi'o version 5.5 where each ring represents an individual genome assembly. After ordering all taxa according to a genomic marker gene tree, genomes were colored following NCBI taxonomy (red: *Lonepinella koalarum*, dark purple: *Pasteurella* spp., light purple: *Haemophilus* spp., pink: *Aggregatibacter* spp., green: *Mannheimia* spp., light green: *Avibacterium paragallinarum*, grey: *Necropsobacter* spp., and yellow: *Rodentibacter* spp. Note, *Actinobacillus* spp. was colored in green here and not orange as in Fig. 1 to show its relation to *Mannheimia* spp. According to GTDB taxonomy, those two genomes are now *Basfia* species. See Discussion section). Each wedge represents a gene cluster. Gene clusters were grouped into mostly shared, shared, private, and in red: exclusively found in *Lonepinella koalarum* genome assemblies: 'LK', and exclusively found in *L. koalarum* strain UCD-LQP1. The gene marker tree was created in PhyloSift version 1.0.1 (*Darling et al., 2014*) with its updated markers database (version 4, posted on 12th of February 2018; *Jospin, 2018*) for the alignment and RAxML version 8.2.10 on the CIPRES web server for the tree inference (*Miller, Pfeiffer & Schwartz, 2010*). Gene clusters were ordered based on presence/absence. Also shown is GC content in light brown, number of genes per kilo base pairs in light grey, number of gene clusters in dark grey, and number of singleton gene clusters in orange, for each assembly, respectively. The heatmap shows ANI (Average nucleotide identity) values > 95%. The ANI heatmap is aligned with the Anvi'o profile, leading to the genome IDs on the y-axis. The Anvi'o database (*Wilkins, 2020g*) and profile (*Wilkins, 2020h*) are accessible on FigShare.

**Table 2 KEGG pathways involved in xenobiotics biodegradation and metabolism.**

| KEGG: Xenobiotics biodegradation and metabolism Pathway | Hits | KEGG ID |
|---|---|---|
| Drug metabolism—other enzymes | 15 | ko00983 |
| Benzoate degradation | 14 | ko00362 |
| Metabolism of xenobiotics by Cytochrome P450 | 9 | ko00980 |
| Chloroalkane and chloroalkene degradation | 8 | ko00625 |
| Aminobenzoate degradation | 6 | ko00627 |
| Xylene degradation | 6 | ko00622 |
| Naphthalene degradation | 6 | ko00626 |
| Dioxin degradation | 5 | ko00621 |
| Chlorocyclohexane and chlorobenzene degradation | 3 | ko00361 |
| Toluene degradation | 2 | ko00623 |
| Nitrotoluene degradation | 2 | ko00633 |
| Styrene degradation | 2 | ko00643 |
| Fluorobenzoate degradation | 1 | ko00364 |
| Ethylbenzene degradation | 0 | ko00642 |
| Atrazine degradation | 0 | ko00791 |
| Caprolactam degradation | 0 | ko00930 |
| Bisphenol degradation | 0 | ko00363 |
| Polycyclic aromatic hydrocarbon degradation | 0 | ko00624 |
| Furfural degradation | 0 | ko00365 |
| Steroid degradation | 0 | ko00984 |

**Note:**
Twenty KEGG pathways known to play a role in plant secondary metabolite degradation (*Kanehisa & Goto, 2000*) were searched in the eggNOG annotations of *Lonepinella koalarum* strain UCD-LQP1 (Hits). The translated amino acid sequences encoded by putative genes in *L. koalarum* can be downloaded from FigShare (*Wilkins, 2020b*).

group but have not been labeled as tannase genes because their sequences are not similar enough to any known tannase genes or because these tannase genes are not annotated in any database. The second largest KEGG pathway was ko00362 benzoate degradation, followed by pathway ko00980 metabolism of xenobiotics by Cytochrome P450 and pathway ko00625 chloroalkane and chloroalkene degradation. KEGG pathways with fewer hits included the degradation compounds such as aminobenzoate, xylene, naphthalene, dioxin, and chlorocyclohexane. Mapping individual genes onto KEGG pathways revealed continuous degradation chains for the following compounds: Azathioprine (pro-drug) to 6-Thioguanine (Fig. S1); Aminobenzoate degradation; that is, 4-Carboxy-2-hydroxymuconate semialdehyde to Pyruvate and Oxaloacetate, which can then be fed into the citric acid cycle (Fig. S2); 2-Aminobenzene-sulfonate to Pyruvate, which, again, can be fed directly into Glycolysis or with another enzyme that was present (1.2.1.10) can be converted into Acetaldehyde, then Acetyl-CoA , and then fed into the Cytrate cycle (Fig. S2). In the group of xenobiotics metabolized by cytochrome P450 there were seven complete chains (Fig. S3): degradation of (i) benzo(a)pyrene, (ii) Aflatoxin B1, (iii) 1-Nitronaphtalene, (iv) 1,1-Dichloro-ethylene, (v) Trichloroethylene, (vi) Bromobenzene, and (vii) 1,2-Dibromoethane. All of these complete, putative conversion chains present

in *L. koalarum* might explain further how this member of the koala gut microbiome contributes to koala gastro-physiology (see discussion below). Amino acid sequences encoded by putative PCD degrading genes in *L. koalarum* strain UCD-LQP1 can be downloaded from FigShare (*Wilkins, 2020b*). A table linking eggNOG annotations to positions in individual assemblies and translated amino acid sequences can be found in Table S2. A complete table of all eggNOG annotations in the assembly of *L. koalarum* strain UCD-LQP1 can be found in Table S3.

*Eucalyptus* spp. leaves contain more than 100 different chemical compounds including phenolics, terpenoids and lipids that are harmful for herbivores, even at low concentration (*Maghsoodlou et al., 2015*). Koalas are highly specialized folivores feeding on these leaves. We assumed that *L. koalarum* plays a beneficial role for koala hosts because some strains have shown experimentally to be able to degrade tannins (*Osawa, 1990*; *Osawa et al., 1995*), and tannic acid was used to isolate *L. koalarum* strain UCD-LQP1. Alas, we did not find any direct evidence for tannase genes in the assembly of *L. koalarum* UCD-LQP1. However, genes encoding several putative pathways involved in plant secondary metabolite degradation were found in the assembly of *L. koalarum* UCD-LQP1. The predicted pathways included those for degradation of compounds that had been extracted from *Eucalyptus* leaves (*e.g.*, benzoate, aminobenzoate and chlorocyclohexane; *Quinlivan et al., 2003*; *Marzoug et al., 2011*; *Sebei et al., 2015*; *Maghsoodlou et al., 2015*; *Shiffman et al., 2017*). Degradation of these PCDs might explain the beneficial role that *L. koalarum* plays in the koala gut microbiome.

## Comparative genomics and unique genes in *L. koalarum*

The GTDB tree clade used to extract related genomes of *L. koalarum* strain UCD-LQP1 consisted mostly of *Haemophilus* spp. ($n = 28$), followed by *Rodentibacter* spp. ($n = 13$), *Pasteurella* spp. ($n = 5$), *Aggregibacter* spp. ($n = 4$), and seven other genera (Table S1). Whole genome marker phylogenetic trees showed that not all genera were monophyletic. This can be seen in Fig. 2 in the way the coloring based on genus name does not group perfectly when taxa are ordered according to their phylogenetic relationship. This was especially the case for *Haemophilus* spp., which is shown in light purple. Some of the *Haemophilus* genomes were grouped together, whereas others grouped with genomes labeled as *Pasteurella* spp., *Necropsobacter* spp. and *Avibacterium* spp. One species of *Rodentibacter* (*R. heylii*) was closest to *Aggregatibacter* spp. (yellow and pink in Fig. 2). *Actinobacillus succinogenes* and *Mannheimia succiniproducens* grouped with *Pasteurella* spp., while the former was the most closely related non-*Lonepinella* genome to *L. koalarum* strain UCD-LQP1. Here it is worth noting that both *A. succinogenes* and *M. succiniproducens* have been renamed in the new GTDB taxonomy to *Basfia succinogenes*, most probably the most closely related taxon to *L. koalarum* that has its genome sequenced to date. Twelve out of the 55 NCBI microbial genome assemblies have different taxonomic names in the new GTDB taxonomy (Table S1). For a discussion of the re-organization and re-naming of the microbial tree of life based on whole genome sequencing see above *Assembly taxonomy and gene annotation*. The whole genome marker gene tree used to order genomes in Anvio's visualization can be downloaded

from FigShare (*Wilkins, 2020c*), as well as its corresponding amino acid alignment (*Wilkins, 2020d*).

Average nucleotide identities have been put forward as a measure of genomic relatedness among bacteria that could help designate genera and be used besides the 16S rRNA gene as a taxonomic marker (*Barco et al., 2020*). Moreover, it has been suggested to use an ANI threshold of larger than 95% to delineate bacterial species (*Goris et al., 2007*). Based on this definition, the genomes used for the comparative genomic analysis with *L. koalarum* are all distinct species (heatmap in Fig. 2). We created a second heatmap visualizing genomic relatedness at the 70% level (Fig. S4). This heatmap revealed several distinct clusters of closely related genomes *vs.* singleton genomes (i.e., taxa that did not group together with anything else at the 70 percent threshold): Cluster (1) *Aggregatibacter* spp., (2) first main *Haemophilus* spp. group, (3) *Rodentibacter* spp., (4) second main *Haemophilus* spp. group, (5) *L. koalarum* genome assemblies and (6) two *Necropsobacter* spp. and another *Haemophilus* spp. Notably, *Rodentibacter heylii*, all *Pasteurella* spp., and *Avibacterium paragallinarum* did not cluster with anything. The heatmap is a way of visualizing sequence similarity groups and overall, it showed that the genera *Haemophilus*, *Pasteurella* and *Rodentibacter* do not represent coherent groups of species or genera. These three genera were found in several sub-groups (clusters in the ANI heatmap in Fig. S4) that have been described previously based on a much larger sample size and a few marker genes (*Naushad et al., 2015*). Even some of the same singleton genomes were reported as their own branches in previous phylogenetic trees (*Christensen et al., 2003*). *L. koalarum* was placed in the middle of a group containing mostly *Haemophilus*, *Pasteurella* and *Basfia* species. Pasteurellaceae, the single constituent family of the order Pasteurellales hosts a diverse group of mostly pathogenic bacteria that had been assigned to this group based on phenotypic traits, often related to their pathology, and GC content (*Mannheim, Pohl & Holländer, 1980*). For example, the genus *Haemophilus* includes a plethora of taxa that cause pneumonia and meningitis in humans, and *Pasteurella* have been associated with a range of infectious diseases in cattle, fowl and pigs (*Naushad et al., 2015*). Moreover, since sequence-based taxonomies have become more common, new genera have been created within each genus, such as for example *Aggregatibacter* (*Norskov-Lauritsen, 2006*) or *Avibacterium* (*Blackall et al., 2005*). We believe that a work-over of the phylogeny and classification of the Pasteurellales is overdue.

The proportion of gene clusters that were unique to the three *L. koalarum* genome assemblies, relative to 55 of their most closely related genomes, was large relative to the size of genes that were unique to other genera in Anvio's pangenome analysis (Fig. 2). There were 282 gene clusters that could exclusively be found in the three *L. koalarum* genome assemblies. Among them, there were 136 gene clusters with complete sequences and COG annotation (Table S4). There were 36 gene clusters unique to *L. koalarum* strain UCD-LQP1 and 19 of these had complete sequences and COG annotations (Table S5).

Out of the 136 gene clusters with known COG functions that were unique to the three *L. koalarum* genome assemblies, 22 different gene clusters fell into the COG category 'Carbohydrate metabolism/transport'. This was the largest category, followed by "Inorganic ion transport" ($n = 15$), "Cell wall", "Transcription" and "Energy production"

**Table 3 Unique gene clusters and their COG IDs in three *Lonepinella koalarum* genome assemblies.**

| Gene ID | COG ID | COG Function | COG Category |
|---------|--------|--------------|--------------|
| 1408 | COG1501 | Alpha-glucosidase, glycosyl hydrolase family GH31 | G |
| 1251 | COG3534 | Alpha-L-arabinofuranosidase\|Alpha-L-arabinofuranosidase | G |
| 1643 | COG2723 | Beta-galactosidase | G |
| 1513 | COG0129 | Dihydroxyacid dehydratase/phosphogluconate dehydratase | G |
| 1404 | COG1349 | DNA-binding transcriptional regulator of sugar metabolism, DeoR/GlpR family | G |
| 1410 | COG2017 | Galactose mutarotase or related enzyme | G |
| 1204 | COG2220 | L-ascorbate metabolism protein UlaG, beta-lactamase superfamily | G |
| 1409 | COG2942 | Mannose or cellobiose epimerase, N-acyl-D-glucosamine 2-epimerase family | G |
| 1250 | COG2211 | Na+/melibiose symporter or related transporter | G |
| 1642 | COG1472 | Periplasmic beta-glucosidase and related glycosidases | G |
| 810 | COG1447 | Phosphotransferase system cellobiose-specific component IIA | G |
| 812 | COG1440 | Phosphotransferase system cellobiose-specific component IIB | G |
| 808 | COG1455 | Phosphotransferase system cellobiose-specific component IIC | G |
| 2231 | COG1263 | Phosphotransferase system IIC components, glucose-specific | G |
| 811 | COG1762 | Phosphotransferase system mannitol/fructose-specific IIA domain (Ntr-type) | G |
| 2230 | COG1621 | Sucrose-6-phosphate hydrolase SacC, GH32 family | G |
| 1372 | COG0524 | Sugar or nucleoside kinase, ribokinase family | G |
| 1407 | COG3684 | Tagatose-1,6-bisphosphate aldolase | G |
| 1729 | COG3711 | Transcriptional antiterminator\|Mannitol/fructose-specific phosphotransferase system, IIA domain | G |
| 1402 | COG1593 | TRAP-type C4-dicarboxylate transport system, large permease component | G |
| 1401 | COG1638 | TRAP-type C4-dicarboxylate transport system, periplasmic component | G |
| 1403 | COG3090 | TRAP-type C4-dicarboxylate transport system, small permease component | G |
| 1023 | COG0859 | ADP-heptose:LPS heptosyltransferase | M |
| 815 | COG3659 | Carbohydrate-selective porin OprB | M |
| 264 | COG3765 | LPS O-antigen chain length determinant protein, WzzB/FepE family | M |
| 1623 | COG1388 | LysM repeat | M |
| 2084 | COG2244 | Membrane protein involved in the export of O-antigen and teichoic acid | M |
| 489 | COG0451 | Nucleoside-diphosphate-sugar epimerase | M |
| 1468 | COG3307 | O-antigen ligase | M |
| 221 | COG3203 | Outer membrane protein (porin) | M |
| 557 | COG1538 | Outer membrane protein TolC | M |
| 199 | COG0810 | Periplasmic protein TonB, links inner and outer membranes | M |
| 1187 | COG2843 | Poly-gamma-glutamate biosynthesis protein CapA/YwtB (capsule formation), metallophosphatase superfamily | M |
| 495 | COG0043 | 3-polyprenyl-4-hydroxybenzoate decarboxylase | H |
| 1371 | COG3201 | Nicotinamide riboside transporter PnuC | H |
| 413 | COG4206 | Outer membrane cobalamin receptor protein | H |
| 817 | COG1477 | Thiamine biosynthesis lipoprotein ApbE | H |
| 2228 | COG2226 | Ubiquinone/menaquinone biosynthesis C-methylase UbiE | H |
| 2071 | COG1401 | 5-methylcytosine-specific restriction endonuclease McrBC, GTP-binding regulatory subunit McrB | V |
| 1160 | COG1132 | ABC-type multidrug transport system, ATPase and permease component | V |

### Table 3 (continued)

| Gene ID | COG ID | COG Function | COG Category |
|---|---|---|---|
| 1058 | COG4823 | Abortive infection bacteriophage resistance protein | V |
| 1573 | COG0251 | Enamine deaminase RidA, house cleaning of reactive enamine intermediates | V |
| 1669 | COG2337 | mRNA-degrading endonuclease, toxin component of the MazEF toxin-antitoxin module | V |
| 2012 | COG0845 | Multidrug efflux pump subunit AcrA (membrane-fusion protein) | V |
| 432 | COG3093 | Plasmid maintenance system antidote protein VapI, contains XRE-type HTH domain | V |
| 1138 | COG2828 | 2-Methylaconitate cis-trans-isomerase PrpF (2-methyl citrate pathway) | C |
| 9 | COG1048 | Aconitase A | C |
| 311 | COG1454 | Alcohol dehydrogenase, class IV | C |
| 1134 | COG3312 | FoF1-type ATP synthase assembly protein I | C |
| 2130 | COG0435 | Glutathionyl-hydroquinone reductase | C |
| 763 | COG0371 | Glycerol dehydrogenase or related enzyme, iron-containing ADH family | C |
| 1584 | COG0778 | Nitroreductase | C |
| 816 | COG1053 | Succinate dehydrogenase/fumarate reductase, flavoprotein subunit | C |
| 514 | COG4972 | Tfp pilus assembly protein, ATPase PilM | W |
| 2098 | COG1116 | ABC-type nitrate/sulfonate/bicarbonate transport system, ATPase component | P |
| 2097 | COG0715 | ABC-type nitrate/sulfonate/bicarbonate transport system, periplasmic component | P |
| 592 | COG0600 | ABC-type nitrate/sulfonate/bicarbonate transport system, permease component | P |
| 818 | COG2807 | Cyanate permease | P |
| 2005 | COG2382 | Enterochelin esterase or related enzyme | P |
| 234 | COG3301 | Formate-dependent nitrite reductase, membrane component NrfD | P |
| 365 | COG3230 | Heme oxygenase | P |
| 1053 | COG0672 | High-affinity Fe2+/Pb2+ permease | P |
| 1051 | COG2822 | Iron uptake system EfeUOB, periplasmic (or lipoprotein) component EfeO/EfeM | P |
| 28 | COG2375 | NADPH-dependent ferric siderophore reductase, contains FAD-binding and SIP domains | P |
| 1158 | COG2223 | Nitrate/nitrite transporter NarK | P |
| 1209 | COG2223 | Nitrate/nitrite transporter NarK | P |
| 1089 | COG4771 | Outer membrane receptor for ferrienterochelin and colicins | P |
| 1052 | COG2837 | Periplasmic deferrochelatase/peroxidase EfeB | P |
| 310 | COG0659 | Sulfate permease or related transporter, MFS superfamily | P |
| 631 | COG4388 | Mu-like prophage I protein | X |
| 662 | COG2932 | Phage repressor protein C, contains Cro/C1-type HTH and peptisase s24 domains | X |
| 2183 | COG5412 | Phage-related protein | X |
| 1780 | COG1943 | REP element-mobilizing transposase RayT | X |
| 1057 | COG2189 | Adenine specific DNA methylase Mod | L |
| 48 | COG1074 | ATP-dependent exoDNAse (exonuclease V) beta subunit (contains helicase and exonuclease domains) | L |
| 47 | COG0507 | ATP-dependent exoDNAse (exonuclease V), alpha subunit, helicase superfamily I | L |
| 2035 | COG3057 | Negative regulator of replication initiation | L |

Note:
A comparative genomic analysis was performed in Anvi'o version 5.5 (*Eren et al., 2015*) to compare three publicly available *Lonepinella koalarum* genome assemblies to 55 closely related genomes. Gene clusters that could only be found in *L. koalarum* relative to the rest were filtered out and annotated with COG (Clusters of Orthologous Groups). Carbohydrate-active enzymes are shaded in grey. Only a subset of COG categories are shown; G: "Carbohydrate metabolism/transport", M: "Cell wall", H: "Coenzyme metabolism", V: "Defense", C: "Energy production", W: "Extracellular", P: "Inorganic ion transport", X: "Prophages and Transposon", L: "Replication and Repair". The complete list of unique gene clusters and their translated amino acid sequence in *L. koalarum* can be found in Tables S4–S6.

($n$ = 11, each) and "Defense" ($n$ = 7; Table 3 and Table S4). The translated amino acid sequences for these gene clusters, extracted from *L. koalarum* strain UCD-LQP1, can be found in Table S6.

Gene clusters in the category "Carbohydrate metabolism and transport" are discussed in detail below. It is worth mentioning that several putative components of the phosphotransferase system were unique to *L. koalarum*. This system transports sugars into bacteria including glucose, mannose, fructose, and cellobiose. It can differ among bacterial species, mirroring the most suitable carbon sources available in the environment where a species evolved (*Tchieu et al., 2001*). *L. koalarum* also stood out in terms of genes coding for cell wall components including for example teichoic acid and other outer membrane proteins (Table 3; Table S4). These outer membrane proteins are diverse and can significantly differ among bacterial species (*Schleifer & Kandler, 1972*). A few other potentially unique gene clusters included genes coding for type IV pilus assembly proteins for species-specific pili and fimbria (*Proft & Baker, 2009*); defense mechanisms, such as putative bacteriophage resistance proteins, phage repressor proteins; and drug transport and efflux pumps. Several of these factors are characteristic for pathogenic bacteria (*Craig, Pique & Tainer, 2004*). Here it is worth noting that a Gram-negative bacterium that was assigned to the genus *Lonepinella* based on 16S rRNA gene sequences caused a human wound infection after a wildlife worker had been bitten by a koala (*Sinclair et al., 2019*).

## Carbohydrate metabolism

Since the majority of unique gene clusters in all three *L. koalarum* genome assemblies were related to carbohydrate metabolism and transport, we decided to screen all three *L. koalarum* assemblies for potential enzymes that assemble, modify, and breakdown oligo- and polysaccharides. Using very stringent selection thresholds of the CAZy database where genes coding for carbohydrate-active enzymes have to be identified by three different methods, we found evidence for the presence of genes encoding 15 different glycoside hydrolase families, three different carbohydrate esterase families, and nine different glycosyltransferase families (Table 4). Note, gene families in *L. koalarum* are predicted to have these activities in carbohydrate metabolism and transport based on characterized other members in the CAZy database, but we do not provide experimental evidence that *L. koalarum* performs these activities. All 28 identified CAZy gene families had also been annotated in the 2,370 eggNOG annotations (Table S3). Glycoside hydrolase families, GH2, GH31, GH32, GH43 and GH77 were only found in the three *L. koalarum* genome assemblies relative to the other taxa in the comparative genomic analysis (see also Table 3 and Table S7). These five glycoside hydrolases are responsible for the hydrolysis of glycosidic bonds. Notably, when *Lonepinella koalarum* was isolated and described the first time as a phylogenetically and phenotypically novel group within the family Pasteurellaceae, enzyme activities were determined using commercially available oxidase/catalase tests as well as high-pressure liquid chromatography (*Osawa et al., 1995*). The new taxon in 1995 (first described *L. koalarum*) showed positive results for beta-galactosidase (putatively enzyme family GH2) and alpha-amylase (putatively enzyme family GH77) and negative

**Table 4 Putative carbohydrate-active enzymes (CAZy) found in *Lonepinella koalarum* genome assemblies.**

| CAZy | Enzyme | LK position |
|---|---|---|
| GH1 | β-Glycosidase; membrane-bound lytic transglycosylase A (MltA) | 1_1622 |
| GH2 | β-Galactosidase | 1_1329 |
| GH3 | Glycoside hydrolase Family 3 | 1_1536 |
| GH4 | α- and β-Glycosidases | 1_1355 |
| GH13 | Major glycoside hydrolase family acting on substrates containing α-glucoside linkages | 1_1263 |
| **GH20** | **Retaining glycoside hydrolases** | **1_757** |
| GH23 | Lytic transglycosylases of GH23 | 1_1073 |
| GH31 | α-Glucosidases | 1_1388 |
| GH32 | Inverting sucrose; invertase | 2_66 |
| GH33 | Glycoside hydrolase family 33 | 1_1214 |
| **GH42** | **Plant cell wall degradation** | **1_1367** |
| GH43 | α-L-Arabinofuranosidase and β-D-xylosidase activity | 1_1369 |
| GH77 | α-Amylase | 3_8 |
| GH102 | Lytic transglycosylases | 1_346 |
| GH103 | Lytic transglycosylase B (MltB) | 2_81 |
| **CE4** | **Deacylation of polysaccharides** | **1_760** |
| CE9 | Deacetylation of N-acetylglucosamine-6-phosphate | 1_1583 |
| CE11 | Carbohydrate esterase family 11 | 1_1738 |
| GT2 | Glycosyltransferase family 2 | 1_759 |
| GT5 | Glycosyltransferase family 5 | 3_4 |
| GT9 | Glycosyltransferase family 9 | 1_999 |
| GT19 | Glycosyltransferase family 19 | 1_1807 |
| GT28 | β-1,4-GlcNAc Transferase | 1_1732 |
| GT30 | Glycosyltransferase family 30 | 1_855 |
| GT35 | Glycogen and starch phosphorylase | 1_1193 |
| GT41 | N-glycosyltransferase | 1_943 |
| GT51 | Murein polymerase | 2_71 |

**Note:**
Genes coding for putative carbohydrate-active enzymes (CAZy enzymes) that were only found in *Lonepinella koalarum* relative to 55 closely related genomes are shaded in grey (corresponding to Table 3), and CAZy enzymes that were only found in the assembly of *L. koalarum* UCD-LQP1 alone are presented in bold. The position of genes potentially coding for these enzymes in the assembly of *L. koalarum* UCD-LQP1 are shown as well. See "Material and Methods" section for details on how assemblies were screened for CAZy enzymes. Positions of these genes in *L. koalarum* genome assemblies ATCC 700131 and DSM 10053 are shown in Table S7.

results for urease, arginine dihydrolase, lysine decarboxylase, and tryptophane desaminase in congruence with the sequence-based results here.

Genes coding for oligosaccharide-degrading enzymes in the families GH1, GH2, GH3, GH42 and GH43 have also been found in another study that was investigating koala and wombat metagenomes (*Shiffman et al., 2017*). Especially GH2, GH3 and GH43 were relatively common in koala metagenomes, relative to wallaby foregut (*Pope et al., 2010*), cow rumen (*Brulc et al., 2009*) and termite hindgut (*He et al., 2013*) metagenomes, where these enzymes had also been characterized. These five glycoside hydrolase families comprise mostly oligosaccharide-degrading enzymes (*Allgaier et al., 2010*); that is, they are

able to break down a specific group of monosaccharide sugars in other bacteria that had been characterized for the CAZy database. However, presumably the major components of koala diet that are difficult to digest for the host are plant secondary metabolites and plant cell walls in *Eucalyptus* leaves, and oligosaccharide-degrading enzymes only play a significant role in a koala's diet after other enzymes have already degraded cellulose in leaf plant cell walls (*Moore et al., 2005*). Oligosaccharides in *Eucalyptus* leaves will be absorbed by the koala in the small intestine and only a small fraction enter the caecum and colon. This means that the bacteria in the hindgut are most likely using their metabolic pathways to process the products of the degradation of complex carbohydrates with cross-feeding among microbiome members. The benefit of this activity to koala nutrition is not well understood. Interestingly, among the genes that code for the three carbohydrate-active enzyme families that were found exclusively in the assembly of *L. koalarum* strain UCD-LQP1, two were actual lignocellulases; that is, microbial enzymes that hydrolyze the beta-1,4 linkages in cellulose (*Allgaier et al., 2010*): Enzyme family GH42 and CE4. GH42 enzymes have mostly been described in cellulose-degrading bacteria, archaea and fungi (*Kosugi, Murashima & Doi, 2002*; *Shipkowski & Brenchley, 2006*; *Di Lauro et al., 2008*). CE4 is a member of the carbohydrate esterase family, which groups enzymes that catalyze the de-acetylation of plant cell wall polysaccharides (*Biely, 2012*). Digestion of plant cell walls, (i.e., cellulose, hemicellulose and lignin), could be a second explanation (besides PCD degradation) of how *L. koalarum* plays a beneficial role in the koala gut microbiome.

## Antibiotic resistance genes

Screening the three *L. koalarum* genome assemblies against the ResFinder database did not result in any detection of antibiotic resistance variants. However, there were three hits in the CARD database. First, all three *L. koalarum* assemblies contained a gene coding for a translated amino acid variant at a specific position (SNP R234F) that had been shown to confer resistance to pulvomycin in other bacterial species based on CARD predictions. Secondly, a variant was found to be encoded in all three *L. koalarum* genome assemblies that had been described before in *Haemophilus influenza* mutant PBP3, conferring resistance to beta-lactam antibiotics (cephalosporin, cephamycin and penam) with SNPs D350N and S357N. The third result was an amino acid position with reference to a protein homolog model in a *Klebsiella pneumoniae* mutant, conferring resistance to the antibiotic efflux pump KpnH (including macrolide antibiotics, fluoroquinolone, aminoglycoside, carbapenem, cephalosporin, penam, and penem). These results are based on predictions from the CARD 2020 database. All three hits are nucleotide sequences in the *L. koalarum* assemblies that are predicted to encode proteins that showed the same amino acid variants as other bacterial species in the CARD database. We do not know whether these variants confer antibiotic resistance in *L. koalarum*. Additional experiments are necessary to confirm that these CARD predictions work for *L. koalarum*. The corresponding nucleotide sequences and CARD output files are deposited on FigShare (UCD-LQP1: *Wilkins, 2020e*; ATCC 700131: *Wilkins, 2020f*; and DSM 10053: *Wilkins, 2020e*).

### Recommendations for future koala management strategies

In previous work, we identified *L. koalarum* as the most predictive taxon of koala survival during antibiotic treatment and we suggested that this bacterium is important for koala health (*Dahlhausen et al., 2018*). Here, we isolated a *L. koalarum* strain from the feces of a healthy koala and sequenced and characterized its genome. We found several putative detoxification pathways in *L. koalarum* strain UCD-LQP1 that could explain its potentially beneficial role in the koala gut for koala survival and fitness. Besides detoxification of plant secondary metabolites, we found several putative genes involved in carbohydrate metabolism, particularly cellulose degradation. Some of these genes were only found in *L. koalarum* assemblies and not in 55 of their closely related genomes. Based on CARD predictions, the *L. koalarum* assemblies contain some sequences that are similar to antibiotic resistance genes in other bacterial species. We suggest confirming these antibiotic resistances in *L. koalarum* experimentally and testing the efficiency of these antibiotic compounds against *Chlamydia* infections in koalas. In light of the various threats that koalas face, from chlamydia infection to wildfires (*Polkinghorne, Hanger & Timms, 2013*), and the growing interest in rescuing and treating them in sanctuaries and zoos, it is important to identify beneficial members of their microbiome. This could (i) help decide which antibiotic compounds to choose during chlamydia treatment in order to maximize persistence of beneficial members in the koala gut microbiome and (ii) guide the development of probiotic cocktails during recovery (*Jin Song et al., 2019*).

## ACKNOWLEDGEMENTS

We thank Céline Caseys ORCID: 0000-0003-4187-9018 for her idea to search eggNOG annotations against KEGG Class 1.11 Xenobiotics biodegradation and metabolism pathways; Chris Brown ORCID: 0000-0002-7758-6447 for his script to download assemblies in batch from GenBank; Cassandra L. Ettinger ORCID: 0000-0001-7334-403X for valuable comments on the pangenome analysis and help with revising the figures. We also thank Dr. Joseph Gillespie ORCID: 0000-0002-5447-7264, Dr. Raphael Eisenhofer ORCID: 0000-0002-3843-0749 and two anonymous reviewers for comments on the submitted manuscript. Isolate 16S Sanger sequencing was performed at the UC Davis College of Biological Sciences DNA Sequencing Facility.

### Funding

Laetitia G.E. Wilkins was supported by the Gordon and Betty Moore Foundation through Grant GBMF5603 (https://doi.org/10.37807/GBMF5603) to Jonathan A. Eisen.
The funders had no role in study design, data collection and analysis, decision to publish, or preparation of the manuscript.

### Grant Disclosures

The following grant information was disclosed by the authors:
Gordon and Betty Moore Foundation: GBMF5603.

## Competing Interests

Jonathan A Eisen is an Academic Editor for PeerJ.

## Author Contributions

- Katherine E. Dahlhausen conceived and designed the experiments, performed the experiments, analyzed the data, authored or reviewed drafts of the paper, and approved the final draft.
- Guillaume Jospin analyzed the data, authored or reviewed drafts of the paper, and approved the final draft.
- David A. Coil conceived and designed the experiments, performed the experiments, authored or reviewed drafts of the paper, and approved the final draft.
- Jonathan A. Eisen conceived and designed the experiments, authored or reviewed drafts of the paper, and approved the final draft.
- Laetitia G.E. Wilkins analyzed the data, prepared figures and/or tables, authored or reviewed drafts of the paper, and approved the final draft.

## Field Study Permissions

The following information was supplied relating to field study approvals (i.e., approving body and any reference numbers):

Jim Nappi and Graham Crawford of the San Francisco Zoo gave permission to collect koala fecal samples.

## DNA Deposition

The following information was supplied regarding the deposition of DNA sequences:

Raw data belonging to *Lonepinella koalarum* strain UCD-LQP1 is available at NCBI BioProject ID PRJNA560698, BioSample; SAMN12598050 and GCF_008723255.1.

## Data Availability

The data are available at this figshare project: https://figshare.com/projects/Lonepinella_koalarum_genome_assembly/74613.

Scripts are also available at Figshare:

Wilkins, Laetitia (2020): Notebook *Lonepinella koalarum* comparative genomic analysis. figshare. Software. DOI 10.6084/m9.figshare.11678262.v1.

## Supplemental Information

Supplemental information for this article can be found online at http://dx.doi.org/10.7717/peerj.10177#supplemental-information.

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
