# Peer review of "Isolation and sequence-based characterization of a koala symbiont: Lonepinella koalarum"

_PeerJ, doi:10.7717/peerj.10177_

## Round 0.1 · original submission · Major Revisions

Dear Dr. Dahlhausen and colleagues:

Thanks for submitting your manuscript to PeerJ. I have now received three independent reviews of your work, and as you will see, the reviewers raised some minor concerns about the research. Despite this, these reviewers are optimistic about your work and the potential impact it will lend to research on koala biology. Thus, I encourage you to revise your manuscript, accordingly, taking into account all of the concerns raised by the reviewers.

While the concerns of the reviewers are relatively minor, this is a major revision to ensure that the original reviewers have a chance to evaluate your responses to their concerns.

The concerns of Reviewer 3 will require the most attention. The reviewer has provided many ideas for improving the structure of your work and delivery of results.

I look forward to seeing your revision, and thanks again for submitting your work to PeerJ.

-joe

·

Basic reporting

The English used throughout this manuscript is clear and the literature cited is relevant.

I would, however, recommend incorporating this recent and highly relevant paper into the introduction (perhaps in the paragraph of lines 62-74):
Faecal inoculations alter the gastrointestinal microbiome and allow dietary expansion in a wild specialist herbivore, the koala (Blyton et al. 2019).

Additionally, on line 80, I would replace Brice 2019 with this reference:
Gut microbes of mammalian herbivores facilitate intake of plant toxins. Kohl KD, Weiss RB, Cox J, Dale C, Dearing MD
Justification being that Kohl et al. 2014 experimentally demonstrated the point you’re making.

In figure 1, I would recommend bolding and placing an asterisk on the UCD-LQP1 leaf for clarity.

In figure 2, I would again recommend annotation of the UCD-LQP1 track to help quickly orient the reader. The ANI heatmap confused for a while until I realised that perhaps it is incorrectly oriented? It makes sense to me for it to be rotated 90 ° counterclockwise.
Actually, I think the issue is that the y-axis of the ANI heatmap is missing from the provided figure (was present when viewing through Anvi’o).

Do the authors know the history of the koala they sampled? When was it brought to the San Francisco Zoo? Was it brought over from a Zoo (or the wild) in Australia? If so, which state/population, etc? What species of Eucalyptus is it being fed? Such information could be useful for future investigations comparing the author’s L. koalarum assembly to ones from different populations.

Overall, the level of reporting and availability of resources in this manuscript is commendable. I love that a Jupyter notebook was uploaded containing extra background and the scripts ran. I can also confirm that the Anvi’o databases they provided are working. However, I could not find the raw sequencing data anywhere. I would request that the authors upload this both for reproducibility, and for future use with improved assembly software/tools.

Experimental design

I see no issues with the methods and analyses used by the authors, and they are cited appropriately.

Validity of the findings

no comment

Additional comments

This paper is a logical extension from the authors’ previous work (Dahlhausen et al., 2018), and I think it’s great to see more work being done on the functional characterisation of host-associated microbes. Overall, I’m happy to recommend acceptance of the manuscript for publication on the provision that my comments are addressed.

Reviewer 2 ·

Basic reporting

The authors isolated a strain of Lonepinella koalarum and investigated genes that may be used for plant secondary metabolite degradation, carbohydrate metabolism, and antibiotic resistance. They compared the isolate genome to two other L. koalarum genomes from Genbank.

The article is well written and includes a sufficient introduction and background. Figures and tables are relevant to the article, appropriately described and labeled. Data is available from NCBI and permits for sample collection is provided. However, the NCBI accession number is missing from the manuscript.

Experimental design

The primary research is within the scope of the journal. A rigorous investigation was performed with well-designed, robust methods that could be reproduced.

Validity of the findings

All underlying data has been provided. Conclusions are well stated, linked to original research question and limited to supporting results.

Additional comments

The fact that you have an isolate means that you could test for antibiotic resistance but it is out of the scope of the manuscript and could be a future project.

The manuscript follows on from previous work that showed that L. koalarum could be used as an indicator of koala health. How common is L. koalarum?

Minor edits are included below:

Line 102: brackets around Phascolarctos cinereus.
Line 389: Supplementary Table S2 doesn’t refer to taxonomic names.
Figure 2: I find ‘private’ confusing. Maybe it would be better to refer to these gene clusters as ‘unique’.
Table 3 title: genome assemblies unitalicised.
Table 4: I’m not sure what the “no approved entry” means. It maybe best to remove it.

Reviewer 3 ·

Basic reporting

Clear and unambiguous English is used throughout and the literature cited is appropriate. The background and context presented in the introduction could be improved (see my point by point responses). The structure is generally appropriate although a section of the results is presented in the conclusions. The article is self-contained.

Experimental design

The research is original, performed to a high technical standard and the methods are well described. The research question is well defined, however, the knowledge gap that it fills could be better expressed. I suggest couching the genetic characterisation of lonepinella around what it tells us about its functional niche and how that related to koala nutrition and digestion. Tannin degradation may be one aspect but it is only a small part.

Validity of the findings

The overall findings are sound. I would, however, suggest a more detailed presentation of some of the outcomes. See my general comments for more details.

Additional comments

The manuscript by Dahlhausen aims to genetically characterise Lonepinella koalarum, a member of the koala gut microbiome that has previously been identified as able to degrade tannin-protein complexes and may improve koala survival during antibiotic treatment. The methods and analyses are appropriate to meet these aims and their presentation is refreshingly detailed. However, I think that the major findings of the study are not well described or elaborated upon.

There is a strong focus on the detection (or the lack there of) of tannin-degrading genes. Yet such genes are poorly characterised and not well represented in annotation databases. Additionally, the benefit of tannin-protein complex degradation to koala nutrition is questionable as it is thought that the liberated protein cannot be absorbed by the koala in the hindgut and may instead only serve to benefit the microbial community. Furthermore, phenotypic studies have already shown that lonepinella is capable of degrading tannin complexes and thus genetic determination of this function provides little additional information about the species.

Instead, I feel that the main benefit of this work would be to identify lonepinella’s functional niche within the microbiome and describe how its functions could benefit koala nutrition, digestion and health. While many of the analyses need to achieve such a characterisation have already been performed, they have not been presented in sufficient detail to build an overall picture of lonepinella’s role in the microbiome. The analyses also mainly describe how lonepinella differs from closely related bacteria but what about its overall all metabolic functions? More details and discussion is needed around specific degradation pathways, what section of these pathways are present in lonepinella and the functions these enzymes and pathways have. The manuscript would also benefit from placing these functions and pathways in the context of koala gastro-physiology and lonepinella’s proposed functional niche within the microbiome. See my point-by point responses for further details.

Point-by Point comments:

Abstract
L21: write species name in full at first mention
L23: What is meant by a “common member”. L. koalarum is found at very low relative abundance in the faecal microbiomes of koalas and is not detected in the majority of koalas (potentially due to it falling below the detection threshold in most cases, see (Alfano 2013, Brice 2019 and Blyton 2019).
L18-34: It would be good if the abstract included a summary of the studies major findings. What unique genes were found? Where did it fall in the phylogenetic tree? What plant secondary metabolite degrading genes were found? etc. It would also be useful to have a conclusion to indicate what these findings suggest for the role of L. koalarum in koala digestion and health.

Introduction
In general the introduction is very focused on PBMCs and their detoxification, however, the koala microbiome as a whole also plays an important role in macro nutrient digestion and fibre degradation (see Brice 2019 and Blyton 2019). This should also be covered in the introduction and be flagged as potential ways that lonepinella could contribute to koala digestion/health. These functions are covered in the manuscript’s analysis of the lonepinella genome and could be nicely given context in the introduction. It would also be useful to indicate why the genomes were screened for genes associated with antibiotic resistance.

L43: They can also cause negative post-digestive feedback by making the koala feel “sick” as discussed in Lawler, Foley and Eschler. Also many of these really are not toxins but rather anti-nutrient compounds. I think it is inappropriate to simply refer to them as toxins. I suggest referring to them as PBMCs.
L70: It is worth briefly outlining what tannin-protein complexes are and how they influence koala nutrition.
L72: You should give a bit more detail on this finding. Was it the abundance of L. koalarum before treatment that was predictive? The presence/absence of the bacteria? Or was it how well L. koalarum was maintained through the treatment that predicted koala survival?
L69-71: I think it is important here to provide a bit more background about what is known about L. koalarum given it is the subject of this study. For example, Osawa showed that L. koalarum may by mucosal associated in the caecum. If doing this extends to introduction too much then some of the detail on the Eucalypt toxins could be condensed.
L80: “well-being” isn’t a scientific term. I suggest revising.

Materials and methods
L207-226: Why was the phylogenetic tree assembled from the 16S gene sequences instead of constructing a full genome tree? What was the benefit of constructing this tree compared with the GTDB tree?

Results and discussion
L289: It is of interest what proportion of isolates from the culturing reported in the methods were found to be L. koalarum by 16s Sanger sequencing and what (if any) other taxa were isolated. Perhaps a short summary could be given in the results?
L290: What was the genome coverage?
L328: How many tannin-degrading genes are present in the database used for the genome annotation? In general tannases are not well characterised. For example, only one KEGG orthalog was present last time I looked.
L331: Was a clear zone present around isolate UCD-LQP1 on the plate? Bacteria without tannin-protein complex degrading capability are capable of growing on those plates if they are tannin tolerant. Only the presence of a clear zone indicates tannin degradation.
L338-340: Again how many tannases are present in the annotation databases?
L346-347: These enzymes can nonetheless play important roles in degradation. The identified enzyme and function of these genes should be summarised in a supplementary table. It would also be worth interrogating these functions and describing them in the text.
L348-349: Or because the tannases are not in the database.
L350-L354: It would be useful to map the identified genes onto these pathways (there are tools for this on the KEGG website) and determine if they form a continuous degradation/synthesis chain (i.e. are all enzymes necessary for conversion between two compounds present?). These are large pathways, what sections of the pathway are represented? What compounds are degraded by the genes identified in L. koalarum? This will give a much more specific indication of what functions lonepinella is performing.
L491: oligosaccharides in Eucalyptus leaves will be absorbed by the koala in the small intestine and only a small fraction would enter the caecum and colon. As you say, this means that the bacteria in the hindgut are most likely using these pathways to process the products of the degradation of complex carbohydrates with cross-feeding among microbiome members. The benefit of such activity to koala nutrition is not well understood.
L492-501: This is an interesting and important finding. Was there evidence that L. koalarum is capable of fibre fermentation and the production of short-chain fatty acids?
L502: reference required

Conclusions
L506-523: These are not conclusions and should be presented as a section on antibiotic resistance in the results section.
L528: This sentence does not fit with the previous sentence as in the first you say beneficial members need to be identified but then explicitly talk about L. koalarum without first justifying that it is one of these beneficial members. I suggest revising.

---

## Round 0.2 · Minor Revisions

Dear Dr. Dahlhausen and colleagues:

Thanks for revising your manuscript. The reviewers are very satisfied with your revision (as am I). Great! However, there are a couple issues to still address and a few minor edits to make. Please address these ASAP so we may move towards acceptance of your work.

Best,

-joe

·

Basic reporting

no comment

Experimental design

no comment

Validity of the findings

no comment

Additional comments

The authors have addressed the reviewer comments and suggestions. I'm happy to recommend the manuscript for acceptance.

I'd like to thank the other reviewers for their valuable feedback and comments.

Reviewer 2 ·

Basic reporting

no comment

Experimental design

no comment

Validity of the findings

no comment

Additional comments

Just some minor comments:

Line 267: 16S rRNA gene sequences
Line 420 and Line 423: Citric acid cycle?
Line 527: phosphotransferase

Reviewer 3 ·

Basic reporting

no comment

Experimental design

no comment

Validity of the findings

no comment

Additional comments

I reviewed a previous version of this manuscript. I find the current version of the maiscript much improved and the authors have adequately addressed my previous concerns. I am particulary pleased to see that they have taken many of my suggestions on board.

I have only a few minor comments/suggestions on the current version and would recommend acceptance of the article subject to them being addressed.

point by point comments:

Introduction
L: 87-90: I think the salient point here is that tannin bound protein is not able to be digested by the koala or utilised by the microbes. Maybe simplify/clarify the current explination?
L94: Well I would assume they were alive at the beginning of the antibiotic treament. I suggest revising this sentence.

Materials and methods
L226: I don’t think the comma is needed in “i.e.,”

Results and discussion
L375: Here and elsewhere in the results and discussion the word toxin is still used when it is probably not the appropriate descriptor. I suggest carefully editing the manuscript to ensure that the word toxin is only used when referring to a compound that has a toxic effect in vivo.

Table 3: This table is quite large and could be moved to the supplimenry material if space limited.

---

## Round 0.3 · accepted · Accept

Dear Dr. Dahlhausen and colleagues:

Thanks for revising your manuscript based on the concerns raised by the reviewers. I now believe that your manuscript is suitable for publication. Congratulations! I look forward to seeing this work in print, and I anticipate it being an important resource for groups studying koala biology. Thanks again for choosing PeerJ to publish such important work.

Best,

-joe